# What Stage Are We at in the Development of Vaccines Against Tick-Borne Diseases?

**DOI:** 10.3390/vaccines13090990

**Published:** 2025-09-22

**Authors:** Weronika Stachera, Magdalena Szuba, Arya Taesung Kim, Subin Yu, Jaeuk Choi, Deborah Nzekea, Yen Ching Wu, Adrianna Brzozowska, Marcin Sota, Marianna Misiak, Monika Dybicz

**Affiliations:** Department of General Biology and Parasitology, Medical University of Warsaw, 02-004 Warsaw, Poland; s082741@student.wum.edu.pl (M.S.); s079366@student.wum.edu.pl (A.T.K.); s079381@student.wum.edu.pl (S.Y.); cjaeuk@khu.ac.kr (J.C.); s087239@student.wum.edu.pl (D.N.); s080830@student.wum.edu.pl (Y.C.W.); s074243@student.wum.edu.pl (A.B.); s087180@student.wum.edu.pl (M.S.); s087165@student.wum.edu.pl (M.M.)

**Keywords:** tick-borne encephalitis, Lyme disease, babesiosis, ehrlichiosis, rickettsiosis, anaplasmosis, vaccines, ticks

## Abstract

The increasing prevalence of Lyme disease, tick-borne encephalitis (TBE), and other tick-borne infections such as *Babesia*, *Ehrlichia*, *Rickettsia*, and *Anaplasma* is a growing public health concern. Existing tick bite prevention strategies are insufficient; therefore, vaccines represent a promising preventive measure. At the moment, only a vaccine for tick-borne encephalitis is available on the market. A vaccine for Lyme disease, however, is at an advanced stage of clinical trials. In this article, we focus on describing the progress in the invention of vaccines for tick-borne diseases. This article analyzes their development and effectiveness.

## 1. Introduction

The increasing prevalence of Lyme disease and other tick-borne infections is a growing public health concern. According to a meta-analysis by Fischhoff et al., a number of factors are associated with an increased risk of tick-borne diseases. These include high deer populations, dense populations of blacklegged tick nymphs, landscapes that combine grassland and forest, low human population density, gardens, cat ownership, and race. Certain landscaping practices, such as brush removal, tree branch trimming, and creating dry barriers between lawns and forests, were associated with an increased risk despite being commonly recommended. More broadly, owning pets also increases the risk of tick bites. The highest bite rates were recorded in children aged five or under, followed by a secondary peak among adults aged 50 to 70 [1]. A meta-analysis by Kassiri et al. found no difference in seasonal tick infestation patterns in humans between the northern and southern hemispheres [2]. Globally, the highest monthly tick bite rates occurred in June, followed by July, August, May and September [3]. Tick-borne diseases pose a significant threat to human health and place a considerable financial burden on healthcare services. For this reason, vaccines appear to offer a promising solution. This article analyzes their development and effectiveness.

## 2. Prevalence of Ticks and Prevention of Tick-Borne Diseases

The spatial distribution of medically important tick vectors in Europe, including *Ixodes ricinus* and *Ixodes persulcatus*, varies widely [3]. The most commonly reported Borrelia species—*B. afzelii*, *B. garinii*, and *B. valaisiana*—are present throughout much of Europe. The highest *I. ricinus* infection rates have been reported in Central European countries [4]. Research by Strnad et al. showed that adult ticks had significantly higher infection rates than nymphs, and females had higher rates than males [5]. Ticks are often infected with more than one pathogen, and several field studies have shown non-random patterns of coinfection [6].

A recently recognized tick-borne illness is caused by *Borrelia miyamotoi*, a spirochete that leads to relapsing fever and is only distantly related to the *Borrelia burgdorferi* sensu lato group responsible for Lyme disease. *B. miyamotoi* was first isolated from *Ixodes persulcatus* ticks in Japan in 1992 [7]. According to the first meta-analysis on the topic, published by Hoornstra et al. in 2022, the prevalence of *B. miyamotoi* in *Ixodes* ticks varied by species and geographic region [8]. This illness is increasingly recognized as a distinct infectious disease, particularly in North America and Asia. In Europe, increased awareness may lead to the identification of more cases. Underreporting is likely due to low awareness in clinical practice and the lack of routine testing. Therefore, *B. miyamotoi* infection should be considered in the differential diagnosis of fever following a tick bite, along with Lyme disease, anaplasmosis, babesiosis, and tick-borne encephalitis [9].

Currently, there is no approved human vaccine for Lyme disease, and diagnosis and treatment are sometimes ineffective. As a result, preventive strategies are a key focus of research. Hinckley et al. conducted a double-blind, randomized, placebo-controlled trial to evaluate whether applying a synthetic acaricide to residential properties would reduce tick encounters and tick-borne diseases [9]. Households in the treatment group received Talstar Professional (FMC, Philadelphia), containing bifenthrin, a synthetic pyrethroid known to reduce *I. scapularis* populations for up to 41 weeks. The treated properties had 63% fewer ticks. However, there was no difference between groups in the number of human tick encounters or disease incidence. This may be due to exposure outside the home, or because human behavior influences risk more than tick density alone [10].

Another study by Mitchell et al. tested the effectiveness of permethrin-treated clothing in preventing blacklegged tick bites [10]. In a two-year randomized, placebo-controlled, double-blind study involving 82 outdoor workers, participants in the treatment group wore factory-treated permethrin clothing. The control group wore identical but untreated garments. Outdoor work hours, tick encounters, and tick bites were recorded weekly. Permethrin-treated clothing reduced tick bites by 65% in the first year and 50% in the second year, resulting in an overall protective effect of 58%. There were no major differences in other preventive behaviors between the groups, and no adverse effects were reported [11].

Harms et al. evaluated the effectiveness of post-exposure prophylaxis after tick bites in a European setting [11]. In an open label randomized controlled trial, they compared a single 200 mg dose of doxycycline within 72 h of tick removal to no treatment. Lyme disease was diagnosed in 10 of 1041 participants (0.96%) in the prophylaxis group, compared to 19 of 648 (2.9%) in the untreated group. This represents a relative risk reduction of 67% (95% CI: 31–84%) and a number needed to treat of 51 (95% CI: 29–180). No serious adverse events were reported. This study, conducted in a primary care setting, supports the use of single-dose doxycycline to prevent Lyme disease after *I. ricinus* bites [12].

Tick-borne relapsing fever (TBRF) is an acute febrile illness. Hasin et al. assessed the safety and effectiveness of post-exposure treatment for TBRF prevention. In Israel, where the study was conducted, TBRF is caused by *Borrelia persica* and transmitted by *Ornithodoros tholozani* ticks. In this double-blind, placebo-controlled study, 93 healthy individuals with suspected exposure were randomly assigned to receive either doxycycline (200 mg on day one, followed by 100 mg daily for four days) or a placebo. TBRF cases were confirmed by fever and a positive blood smear. All 10 confirmed cases occurred in the placebo group. These results suggest that doxycycline is 100% effective in preventing TBRF after potential exposure. The study concluded that doxycycline is both safe and effective in high-risk settings [12].

Educational interventions in schools play a critical role in prevention, as children are especially vulnerable to tick bites. Increasing awareness of tick-borne diseases and preventive behaviors is therefore essential.

## 3. Tick-Borne Encephalitis

Tick-borne encephalitis (TBE) is a viral infection of the central nervous system caused by three flavivirus subtypes: the Siberian, Far Eastern, and European. The disease is endemic in many parts of Europe and Asia and is transmitted by ticks from the genus *Ixodes* [13]. TBE is most commonly contracted through a tick bite. However, it is also possible to become infected by consuming unpasteurized dairy products [14].

Tick-borne encephalitis (TBE) can have serious health consequences, so it is important to raise awareness of how the disease manifests and how it can be prevented. Many patients experience a biphasic course of illness. The first phase is nonspecific and is characterized by fever, fatigue, headache, and muscle pain. It typically lasts around four days. This is usually followed by a one-week period of improvement or an asymptomatic. The second phase develops as meningitis, meningoencephalitis, or meningoencephalomyelitis [13].

Although TBE can be prevented by vaccination, which is the most efficacious safeguard against this viral infection, the incidence of the disease is still increasing [13,15]. Currently, there are five vaccines against TBE available on the market [16]. However, only two of them are available in Europe: FSME-IMMUN^®^ (Pfizer Europe, Brussels, Belgium) and Encepur^®^ (Bavarian Nordic, Hellerup, Denmark). Both of them contain a formalin-inactivated European subtype of the TBE virus (TBEV). FSME-IMMUN^®^ also induces cross-reactive immunological responses against other subtypes. And both are also registered for children, but the pediatric dose contains 50% of the antigen of the regular dose [15].

Whole viruses that have been chemically inactivated using formaldehyde are used to produce some vaccines. This method of inactivation is well-known and straightforward, but the process itself has several disadvantages: for example, the chemical is toxic to the environment and to humans, and formaldehyde affects the antigenicity of vaccines. An alternative method of pathogen inactivation is ionizing radiation, which damages nucleic acids while leaving other structural components largely intact. Finkensieper et al. propose using low-energy electron irradiation to effectively inactivate viruses while preserving other structural components. However, this method is not yet used to produce a TBE vaccine [17].

The standard vaccination schedule includes three doses, followed by a booster within three years of completing the primary series. Subsequent boosters are usually advised every three to five years, depending on the patient’s age. In a systematic review of the literature, Frederick J. Angulo et al. write that in immunocompetent patients, humoral and cellular responses after a booster dose persist longer than previously thought for both FSME-IMMUN and Encepur vaccines. The article also suggests that the time between booster doses could be extended to 10 years, as this would have a positive overall impact on public health by encouraging more people to get vaccinated. Some countries, Switzerland and Finland, for example, are already recommending a 10-year booster interval [15].

There have been reports of TBE cases despite patients having received all doses of the vaccine. This inadequate protection against infection, as well as the high price of existing vaccines, makes the search for new, cheaper vaccines urgent. A new promising approach is to use virus-like particles (VLPs) based on recombinant proteins that are structurally very similar to natural virions. These particles are produced spontaneously during flavivirus infection and can also be produced using eukaryotic platforms. Marta Zimna et al. proposed a new VLPs vaccine candidate produced using an unconventional *Leishmania tarentolae* expression system. This approach offers several advantages. Since there is no need to work with viruses, VLPs are safer during the production process. *L. tarentolae* cultures can easily be scaled up for industrial-scale production. This expression system is significantly cheaper than producing inactivated vaccines. Furthermore, this vaccine could be potentially used to immunize reservoir animals, thereby reducing viral transmission to humans. The vaccine has already been tested on mice and proven effective. No side effects were observed during vaccination [16]. The same group of researchers is currently working further on this vaccine. In 2024, they published a paper on the efficacy of the vaccine when using different adjuvants and various routes of administration [18]. Attempts to create a vaccine with VLPs were also made by Jielin Tang et al. They described an efficient construction of TBE VLPs by plasmid-driven transfection of viral proteins in mammalian cells. The results of experiments on mice prove that the vaccine has three main effects: it stimulates a response in CD4+ T cells that produces multiple cytokines, it induces a response in the form of neutralizing antibodies, and it protects mice against a lethal TBEV infection [19].

Another approach to creating a vaccine is to use the Langat virus (LGTV), which is a naturally attenuated member of the tick-borne encephalitis virus serocomplex. It is endemic in Southeast Asia, but no cases of LGTV-associated human disease have ever been reported. Based on animal studies, LGTV is known to exhibit lower neurovirulence and neuroinvasiveness compared to TBEV. Konstantin A. Tsetsarkin et al. generated and characterized chimeric TBEV/LGTV strains containing targets for miRNAs expressed in the central nervous system, which were inserted at two distant locations in the viral genome. The most promising TBEV vaccine candidate is the miRNA-targeted virus T/1674-mirV2, but further investigation is required [20].

## 4. Lyme Disease

### 4.1. Introduction

Lyme borreliosis is a zoonotic infection caused by spirochetes of the *Borrelia burgdorferi* sensu lato complex. It is the most frequently reported vector-borne disease in North America and Europe, and recent estimates suggest that approximately 476,000 people are diagnosed and treated for Lyme disease each year in the United States, with over 200,000 additional cases annually in Europe [21,22]. Because present-day prevention still relies on tick avoidance, prompt tick removal, and early antibiotic therapy, the withdrawal of the only licensed human Lyme vaccine in 2002 left a conspicuous gap in public health tools [23].

Over the last two decades, understanding of *Borrelia*’s antigenic variation and immune evasion has deepened, and vaccine platform technologies have matured [24]. Consequently, multiple vaccination strategies—including subunit, vector-targeted, and mRNA-based approaches—are now under evaluation [25]. The most advanced candidates once again focus on outer surface protein A (OspA)—a lipoprotein abundantly displayed in the tick midgut, which, when targeted by pre-existing antibodies, can block *Borrelia* transmission to the human host [26]. Parallel work has explored outer surface protein C (OspC), complement regulator-acquiring surface protein Z (CspZ), and various borrelial or tick salivary antigens that might broaden strain coverage or interrupt the enzootic cycle [27,28,29]. Still other groups have piloted reservoir-targeted oral vaccines and anti-tick immunizations that, in theory, could suppress community transmission rather than merely protect individual recipients [30,31].

### 4.2. Historical Vaccine Efforts and Lessons Learned

The concept of vaccinating against Lyme disease emerged in the 1990s after demonstrations that *Borrelia* outer surface proteins could induce protective immunity in animal models [32,33]. Among these antigens, OspA was identified as a particularly promising target. OspA is strongly expressed by *Borrelia* in the tick midgut during feeding, and antibodies against tbevOspA can neutralize the spirochetes before they ever enter the human host [34]. This transmission-blocking mechanism was leveraged in the first-generation Lyme vaccines developed in the late 1990s [35]. SmithKline’s LYMErix^®^ vaccine—containing a lipidated full-length OspA (serotype 1) adjuvanted with alum—became the first (and so far, only) licensed human Lyme vaccine [34]. In Phase III trials, LYMErix showed moderate efficacy: roughly 49% protection after two doses, rising to 76% after the full three-dose series [35]. Notably, vaccination not only reduced symptomatic Lyme disease but also prevented asymptomatic *Borrelia* infections, as evidenced by the absence of seroconversion [34,35]. These efficacy outcomes—comparable to those of other tick-borne disease vaccines—suggested that OspA immunization could significantly reduce Lyme risk in endemic areas. The vaccine was licensed in 1998, and about 1.5 million doses were distributed in the United States during its initial years of use [36].

LYMErix’s withdrawal a few years post-licensure was not due to any definitive safety failure but rather to a convergence of public perception issues and market forces [23,36]. Extensive post-marketing surveillance and retrospective analyses found no significant difference in chronic joint symptoms (including arthritis) between vaccine and placebo recipients [36]. However, media reports and a class-action lawsuit alleging vaccine-induced autoimmunity drove negative public sentiment [23]. At the same time, Lyme disease incidence is regionally focal, and many at-risk individuals were skeptical of or unfamiliar with the vaccine, leading to lower-than-expected uptake [36]. Caught in this climate of hesitancy, the manufacturer voluntarily withdrew LYMErix in 2002, and development of other human Lyme vaccines stalled [36]. This episode underscored the importance of transparent safety communication and the need to proactively address public concerns in any future vaccine rollout [23]. In parallel with LYMErix, a second OspA-based vaccine candidate called ImuLyme^TM^ was developed by Pasteur Mérieux Connaught. ImuLyme used a similar recombinant OspA antigen with the same 0-, 1-, 12-month schedule, and its Phase III trial showed 68% efficacy after two doses, rising to 92% one year after the third dose—indicating a robust booster response [36]. Notably, ImuLyme’s clinical testing detected no arthritis signal, and most adverse events were limited to mild injection site or systemic reactions [36]. Despite this favorable profile, ImuLyme was never marketed: the sponsor opted to halt its launch following the market failure of LYMErix [23]. The withdrawal of these first-generation vaccines underscored the critical importance of public acceptance [23].

Importantly, the scientific legacy of LYMErix and ImuLyme has guided next-generation efforts. Both vaccines provided proof-of-concept that OspA-based immunization in humans can confer meaningful protection against Lyme disease [26]. They also highlighted the necessity of addressing *Borrelia*’s strain diversity. LYMErix, containing OspA from only a U.S. strain, was less applicable to Eurasia, where multiple OspA serotypes circulate [37]. This realization spurred the development of multivalent OspA vaccines incorporating the major serotypes to ensure global utility [38]. Additionally, the continued use of OspA vaccines in dogs for over two decades—without incident and with clear protective benefits—has provided reassurance about the approach’s safety and durability in a veterinary context [39]. Collectively, these experiences set the stage for a renewed pursuit of Lyme vaccines, now with improved antigen designs and a more cautious engagement of public stakeholders [40].

In addition to the OspA-based clinical candidates, a range of preclinical antigen targets has also been explored [41]. Among these, OspC—a highly variable surface protein expressed during the early stage of mammalian infection—has shown strong immunogenic potential, prompting efforts to overcome its variability. Chimeric subunit vaccines containing epitopes from multiple OspC variants have indeed induced broad antibody responses in murine models [27]. Other antigens such as OspB, OspD, and BmpA (P39) elicited antibody responses but ultimately failed to provide robust protection in challenge studies [32]. Notably, the factor H-binding protein CspZ and the antigenically variable VlsE have demonstrated encouraging bactericidal activity and even conferred protection in certain mouse studies [28,41], while novel targets like BBK32, BBA52, and BB0405 act during the tick infection phase and may complement existing strategies by blocking transmission at the vector stage [42,43]. However, the protective efficacy of VlsE-based vaccination remains uncertain; for example, one study found that the conserved region of VlsE did not generate protective immunity [44], so this particular claim may need further verification. These various research efforts highlight the complexity of Lyme vaccine development, where antigenic diversity, immune evasion mechanisms, and vector–host–pathogen interactions must all be addressed [24,25]. Despite past setbacks, the early vaccine candidates provided critical insights and essentially laid the groundwork for more advanced multivalent formulations like VLA15, now in late-stage trials [40].

Table 1 summarizes key features of the historical vaccines of Lyme disease alongside subsequent candidates with the evolution of strategies and outcomes. In the next section, we examine how OspA-based vaccine design has progressed since the LYMErix era—leading to a new multivalent OspA vaccine candidate that has reached Phase III trials (VLA15)—and how this approach aims to overcome past limitations.

### 4.3. The Return of OspA—From Monovalent Prototypes to the Multivalent VLA15

The shortcomings of first-generation, single serotype OspA vaccines prompted a redesign that would (I) broaden strain coverage, (II) remove the human LFA1-like sequence that had fueled autoimmune speculation, and (III) shorten the primary series while preserving immunogenicity [23,25]. VLA15 meets those criteria by fusing the C-terminal halves of six OspA serotypes (ST 16) into three lipidated heterodimers, each adsorbed to aluminum hydroxide [38,49]. Lipidation restores the natural N-terminal acylation that augments toll-like receptor signaling, whereas the C-terminal truncation deletes the disputed LFA1-overlapping epitope without disturbing conformational B cell epitopes [37,50].

An overview of the clinical development of the multivalent OspA vaccine VLA15, as described below, is shown in Table 2.

In preclinical studies, a three-dose series of VLA15 over a 0–2–4 week schedule protected mice (and guinea pigs) from challenge with *Borrelia burgdorferi* sensu stricto (OspA serotype 1), *B. afzelii* (ST2), *B. bavariensis* (ST4), and two *B. garinii* strains (ST5/6); additionally, vaccine-induced antibodies showed functional growth inhibition against the tick-refractory serotype 3 strain when tested using chicken complement as a co-factor. A booster given at five months in these animal models further elevated antibody titers and roughly doubled their half-life, foreshadowing the need for periodic boosting to maintain immunity [38].

Phase I (n = 179, 18–39 years) demonstrated that a three-injection primary series (0–1–2 months) was well tolerated, with mostly mild reactogenicity and no vaccine-related serious adverse events reported. The vaccine elicited robust anti-OspA antibody responses to all six serotypes; however, titers declined back to near baseline by ~11 months post-vaccination. Encouragingly, a booster dose at month 13 induced a strong anamnestic response (restoring high antibody levels), although the post-booster decline in titers paralleled that seen after the initial series—indicating that long-term protection will likely require scheduled boosters [49].

Phase II optimization (two observer-blinded trials; n = 821): the VLA15 regimen was refined. A 180 µg dose administered at 0, 2, and 6 months produced higher peak geometric mean titers and slower antibody waning than either a lower 135 µg dose or a more accelerated 0- and 2-month schedule. Local injection-site pain/tenderness was the most common adverse event, whereas systemic symptoms (e.g., fatigue, headache) occurred at frequencies comparable to placebo and were generally mild [51]. A Phase II extension trial evaluating an 18-month booster showed a 40- to 60-fold increase in antibody titers across all OspA serotypes, and these boosted titers remained significantly higher than in placebo recipients at month 30, with no new safety signals observed [52].

A Phase III trial (VALOR) is currently in progress, enrolling approximately 6400 participants ≥ 5 years of age, who receive VLA15 (180 µg) on a 0, 2, and 5–9-month schedule followed by a booster ~12 months later. This pivotal study will assess efficacy (prevention of laboratory-confirmed Lyme disease cases), monitor safety and lot consistency, and explore an immunological correlate of protection. Primary completion is anticipated in late 2025, with regulatory submissions planned for 2026 if prespecified efficacy endpoints are met [53]. Taken together, VLA15 illustrates how rational antigen engineering—guided by lessons from the LYMErix era—can expand strain coverage while maintaining a favorable safety profile in over 1000 human subjects. The chief remaining challenge is the rapid waning of antibody levels, but this appears manageable with annual booster doses.

Importantly, establishing a robust correlate of protection will be critical to inform optimal booster timing and to benchmark future Lyme vaccine candidates [51,52].

### 4.4. Beyond OspA—Complementary Antigen Strategies

Although an OspA backbone remains the most clinically advanced approach, several ancillary antigens and delivery concepts have shown sufficient promise to merit inclusion in a future, layered vaccine architecture [27].

OspC is upregulated in *Borrelia* immediately after a tick begins feeding. Antibodies against certain hypervariable surface loops of OspC can neutralize spirochetes early, before they disseminate in the host. To broaden coverage across OspC variants, researchers have created chimeric “mosaic” proteins that graft multiple OspC loop epitopes onto a single scaffold. These chimeritopes elicit bactericidal antibodies against diverse strains in mice. However, full protection in animal models has only been achieved when an OspC chimeritope is co-administered with an OspA antigen, underscoring that OspC-based components are best used as a complement to OspA rather than as standalone vaccines [27].

CspZ is a *Borrelia* outer-surface protein that binds to human factor H, helping the spirochete evade complement attack. A “YA” double mutant of CspZ (with I183Y and C187S substitutions) was engineered to abolish factor H binding while preserving the protein’s native fold, thereby exposing its protective epitopes. In a murine model, two intramuscular doses of this CspZ-YA variant (either I183Y or C187S) generated robust borreliacidal antibody titers and completely prevented Lyme infection—spirochetes were eradicated from tissues and Lyme arthritis was avoided. This highlights CspZ-YA as a strong adjunct candidate for a multi-component vaccine [54].

Fine-mapping studies of OspA have identified a 20-amino acid segment (residues 221–240) that is highly conserved among *B. burgdorferi* sensu stricto isolates and even shares a closely related sequence in OspB. Antiserum raised against a peptide from this region showed potent complement-dependent killing of *B. burgdorferi*—approximately 86% of strain B31 spirochetes were killed in vitro [55]. This conserved linear epitope also appeared to target OspB-expressing bacteria (due to OspA/OspB sequence homology). Such findings suggest that a short peptide representing this epitope could substitute for the full-length OspA in a future multi-epitope vaccine design [48].

FtlA and FtlB are paralogous lipoproteins expressed by *Borrelia* during the tick’s blood meal (the “feeding tick larval” phase). These antigens are thought to be involved in the spirochete’s survival or transmission in the vector. Antibodies raised against FtlA or FtlB can kill a significant proportion of *B. burgdorferi* in culture, on the order of 65% or more of spirochetes in vitro. Because FtlA/B are found across many *Borrelia* genotypes and are expressed in the tick, an FtlA/FtlB-based vaccine component could act on the pathogen within the tick vector, potentially reducing transmission regardless of the infecting *Borrelia* strain [48].

Another innovative concept is to target tick proteins or combine protection against multiple tick-borne diseases. Anti-tick vaccines would induce immunity against tick salivary or midgut proteins, impairing the tick’s ability to feed or transmit pathogens. Additionally, a single vaccine could be formulated to protect against both Lyme disease (bacterial *Borrelia* infection) and tick-borne encephalitis (TBE, a viral infection endemic in parts of Europe). Modeling from Slovenia suggests that a dual Lyme/TBE vaccine could be cost-effective if its price per dose is kept at approximately EUR 55 or below, assuming it has efficacy comparable to existing separate Lyme and TBE vaccines. A Markov health-economic model predicted an acceptable cost-effectiveness ratio at that price point, meaning such a combined vaccine would provide good value in areas where both diseases co-exist, given similar effectiveness to the single-pathogen products. This dual approach (or even an anti-tick approach) could offer broader protection with one shot, essentially blunting multiple tick-borne threats simultaneously. Importantly, to maximize the efficacy of anti-tick vaccines, vaccination schedules may need to be aligned with the seasonal emergence of the relevant tick feeding stages [31].

Collectively, these strategies illustrate that Lyme disease prevention is likely to evolve beyond a single-antigen vaccine. OspA will remain a central component (given its proven ability to target spirochetes in the tick midgut), but it can be bolstered by additional antigens targeting different stages of the spirochete’s life cycle. By including antigens like OspC or CspZ that act in the host’s early infection phase, or vector-based targets like Ftl proteins or reservoir-oriented vaccines, a next-generation Lyme vaccine could provide more comprehensive protection. The trend is toward polyvalent or “ecosystem-based” solutions in which multiple points in the transmission cycle are interrupted.

### 4.5. Immunological Considerations and Correlates of Protection

Unlike many viral vaccines that have a well-defined neutralizing-antibody threshold, Lyme borreliosis currently lacks an agreed serological correlate of protection. Early LYMErix trials did provide some insight: analyses suggested that a serum anti-OspA IgG concentration of roughly ≥6 µg/mL was sufficient to block tick-to-host transmission of *Borrelia*, whereas a much higher titer on the order of ≥213 µg/mL might be required to completely eradicate *Borrelia* in the tick midgut [41]. In other words, relatively modest anti-OspA levels can prevent the spirochete from moving into the host, but extremely high levels are needed to clear every *Borrelia* from the feeding tick. Because of the absence of a simple threshold level that guarantees protection, current vaccine studies rely on other measures. For instance, trials of the second-generation Lyme vaccine VLA15 have focused on immunological readouts like the geometric mean fold rise (GMFR) in antibody titers and on functional assays such as the serum bactericidal assay (SBA) as surrogate markers of protection. It is expected that the ongoing Phase III trial (VALOR) will include an immunology sub-study to help identify an official correlate of protection. Establishing such a correlate will be essential for guiding booster schedules and could enable accelerated approval of improved or follow-on products in the future. From a public health perspective, waning antibody levels mean that booster doses will likely be required to sustain protection over time. Clear guidance on booster schedules will therefore be critical not only for maintaining individual immunity, but also for ensuring confidence in large-scale vaccination programs [41,53].

The short half-life of anti-OspA IgG (≈4–6 weeks) observed in both first- and second-generation vaccines suggests that annual boosters, timed before peak tick activity, will be required. Memory B cell data are limited, but murine findings indicate that lipidated OspA constructs can generate recall responses exceeding primary peaks, supporting a boost and maintain paradigm similar to seasonal influenza vaccination [53].

### 4.6. Remaining Gaps and Future Directions

The clinical successes obtained so far are encouraging, yet several knowledge voids must be bridged before mass implementation can be contemplated.

#### 4.6.1. Age-Specific Evidence

The VALOR protocol enrolls participants as young as five years, yet children below that threshold and adults above 65 remain largely unstudied [34]. Because immunosenescence and immune ontogeny can both modulate antibody quality as well as reactogenicity, it is conceivable that dosing, booster cadence, or even adjuvant selection will need to diverge across age strata. Targeted trials in these extremes of age, followed by post-authorization pharmacovigilance, will therefore be indispensable.

#### 4.6.2. Co-Endemic Pathogens

*Ixodes* ticks are competent vectors not only for *Borrelia* but also for *Anaplasma phagocytophilum, Babesia microti*, and Powassan virus, among others [34]. A polyvalent product—whether achieved through antigen coformulation, antitick salivary targets, or sequential combination schedules—could plausibly deliver broader clinical value than an OspA-only vaccine. Demonstrating additive or synergistic immunity, while preserving safety, represents a pivotal next step for developers [56].

#### 4.6.3. Public Confidence and Communication

The withdrawal of LYMErix illustrated that regulatory approval and scientific rigor do not, in themselves, translate into uptake. Stakeholder engagement that includes primary care physicians, Lyme advocacy groups, and social media influencers—before licensure—may help neutralize misinformation and set realistic expectations regarding booster needs and the spectrum of protection [57].

#### 4.6.4. Geographical Antigen Breadth

Seroepidemiological surveys have revealed Asian lineages such as *B. valaisiana* and *B. spielmanii* that fall outside the current six-serotype VLA15 panel. Ongoing genomic surveillance should guide whether additional C-terminal OspA fragments, or entirely distinct surface proteins, ought to be incorporated into future iterations so that vaccine utility can extend beyond North America and Europe [36].

#### 4.6.5. Correlates of Protection

Finally, although high anti-OspA titers correlate with reduced transmission in animal models, a universally accepted serological cutoff has yet to be defined [35,41]. The immunology sub-study within VALOR is expected to designate such a threshold; its validation will enable immunobridging studies and streamline the licensure of next-generation combinations.

Addressing these gaps in a deliberate, data-driven manner will determine whether the current wave of Lyme disease vaccines fulfills its promise of durable, population-level impact.

### 4.7. Conclusion

The contemporary Lyme vaccine pipeline reflects a maturing field that has assimilated the technical and social lessons of the 1990s. VLA15 now stands at the threshold of efficacy confirmation and, if successful, could restore a preventive option lost two decades ago. Meanwhile, second-wave antigens such as OspC chimeritopes and CspZ mutants offer routes to broaden mechanistic coverage, and reservoir or vector-directed approaches hold promise for community-level impact. Establishing robust correlates of protection and cultivating public trust will be as decisive as any molecular innovation. Should these elements converge, the next generation of Lyme vaccines may finally deliver durable control of a disease that continues to advance geographically and numerically.

## 5. Babesia

### 5.1. Introduction

Babesiosis is a tick-borne parasitic disease caused by the genus *Babesia*. It can infect both humans and animals. There are many types of *Babesia* species which cause human Babesiosis: *Babesia microti*, mainly found in the US; *Babesia divergens*, which is commonly found in Europe; as well as *Babesia duncani*, and *Babesia venatorum*. The transmission of this disease is mainly through tick bites, as well as through contaminated blood transfusions between people, and the transplacental route during pregnancy. Babesiosis has a peak prevalence time during spring and summer, when nymph ticks are highly active [58,59].

*Babesia* is an intra-erythrocytic protozoan parasite. Its life cycle is highly dependent on red blood cells, with vertebrate animals being one of the targeted hosts. Once the human host becomes infected with *Babesia*, the symptoms usually develop after 1–4 weeks. The person may be asymptomatic or present with flu-like symptoms, such as coughing, fatigue, and muscle aches. However, Babesiosis can lead to severe complications, especially in immunocompromised or asplenic people. Complications include hemolytic anemia, disseminated intravascular coagulation (DIC), acute respiratory distress syndrome (ARDS), and organ failure. This shows the importance of investing in vaccines for Babesiosis [58,59].

In this part of the review, we will mainly focus on vaccinations for *B. microti* and possible recombinant subunit vaccinations for *B. bovis* and *B. divergens.*

### 5.2. Development and History of Babesia Vaccines

Following the discovery of *Babesia*, researchers in the early 1990s used blood from recovered animals as a form of infection-derived protection, forming the basis for modern live attenuated vaccines. Between the 1950s and 1970s, *B. bovis* and *B. bigemina* strains were used for live attenuated vaccines in serial passage in splenectomized cattle [58,59].

By the late 1980s–1990s, focus shifted to identifying immunodominant surface proteins. In the 2000s–2010s, DNA vaccines encoding *Babesia* antigens were tested in animals but never widely adopted. This led to recombinant protein vaccines using glycosyl-phosphatidyl-inositol (GPI)-anchored proteins, with subunit trials—like the Bd37 antigen from *B. divergens*—showing partial to full protection in cattle when paired with strong adjuvants.

Since the 2010s, preclinical human studies have explored liposomal whole-parasite vaccines and other subunits [60].

### 5.3. Vaccines for Babesia Microti

Currently, there are a few types of vaccines against *B. microti* that are still under development, which have the potential to be developed and used in future clinical trials.

#### 5.3.1. Whole-Parasite Vaccines

A characteristic of a whole-parasite vaccine is that it introduces the entire parasite organism, either in an attenuated form or an inactivated form. This method presents the immune system with a broad spectrum of parasite antigens that could mimic natural infection, potentially decreasing the immunity from lack of variant recognition, and helps with boosting immune effectiveness in response to infection [61,62].

In attenuated vaccines, although the parasite is weakened, the efficacy of the immunity is not impacted. During the experiment, tafuramycin-A, a synthetic analog that interferes with the parasite’s DNA, resulting in inhibition of replication, was injected into the mice; the result shows the development of immune protection without parasitemia presenting. Additionally, the safety of the attenuated vaccine has been observed in severe combined immunodeficient mice, with no infection or complication findings [61,63].

Inactivated vaccines using killed parasites enclosed in liposomes have shown protective effects in mice, including in asplenic ones, with lyophilization preserving efficacy. However, only CD4+ T cells and macrophages mediated the response, with minimal B cell and antibody involvement. This concerns the need for strategies that can enhance antibody responses [61].

While the whole-parasite vaccines present certain potential advantages and beneficial aspects, several significant limitations should be taken into consideration. This includes the risk of contamination during vaccine production, the inability to maintain cultures of *B. microti*, and concerns about immunogenicity. Inactivated vaccines may require potent human-specific adjuvants, while attenuated ones risk reversion to virulence or residual pathogenicity in immunocompromised individuals. Furthermore, the presence of RBC membranes in such vaccines raises the possibility of triggering autoimmune responses against host RBCs [61].

#### 5.3.2. Subunit Vaccines

Compared to whole-parasite vaccines, which could present a larger spectrum of antigen phenotype recognition, subunit vaccines mainly target a specific component of the parasite. There are various subunit vaccines aimed at different parts, e.g., surface antigens, internal antigens, moving junction proteins, and cytoadherence-blocking antigens. In this section of the review, we will mainly focus on the surface antigen subunit vaccines [61].

Surface antigens are the most ideal prototype for subunit vaccines due to their easy accessibility to immune cells and antibodies, facilitating parasite elimination. *Babesia microti* Surface Antigen 1 (BmSA1) is a strong candidate for this approach [61].

BmSA1 is a GPI-anchored protein that is abundantly expressed in plasma merozoites of *Babesia microti* and has the ability to bind red blood cells, potentially contributing to erythrocyte invasion. It is highly conserved across *B. microti* strains and strongly immunogenic in humans. The structure of BmSA1 is very similar to the Bd37 protein of *B. divergens*, featuring an N-terminal disordered region and a central alpha-helical domain. Uniquely, BmSA1 is secreted in both soluble and vesicle-bound forms during infection, indicating a likely role in immune modulation rather than direct erythrocyte invasion. Its high expression levels make it a strong candidate for diagnostic use [61,63].

In vitro studies have demonstrated that antibodies targeting BmSA1 can inhibit parasite growth and block red blood cell adhesion, highlighting its potential as a vaccine antigen. However, when BmSA1 was formulated with alhydrogel adjuvant (a human-approved aluminum-based adjuvant used to enhance the immune response), it did not reduce the severity of *Babesia* infection in vaccinated mice [61,63].

### 5.4. Vaccines for Other Babesia Species

From the 1980s, babesiosis vaccine development shifted from live attenuated vaccines toward safer and more effective options, focusing on the following: (I) killed parasites, (II) soluble parasite antigens (SPAs), and (III) recombinant parasite proteins [64].

### 5.5. Vaccines for Babesia Bovis

In Australia, vaccines using killed parasite extracts showed protection levels comparable to live vaccines against *B. bovis*. Subsequent cattle trials identified three protective antigens (12D3, 11C5, and 21B4). Around the same time, SPAs protected dogs and rats against major *Babesia* species, supporting further SPAs vaccine development [64]. SPAs from various *Babesia* species induce immune responses that reduce clinical disease symptoms rather than directly affecting the parasite [65]. However, protection against *B. bovis* varied geographically, ranging from no protection to high efficacy. Differences were also noted between homologous and heterologous challenge outcomes, raising concerns about a universal vaccine’s effectiveness in endemic areas [64].

### 5.6. Vaccines for B. divergens

*B. divergens* is the main cause of bovine babesiosis in Europe and can cause fatal human infections. Studies identified Bd37, a ~37 kDa molecular weight GPI-anchored immunodominant antigen (the immune system strongly recognizes and targets during parasite exposure), as a key vaccine target [66,67]. Purification of Bd37 from in vitro culture supernatants yielded several fractions, including fraction 4 (F4), used in vaccine trials. After the purified Bd37 from in vitro culture supernatants and characterization, a monoclonal antibody (mAb) F4.2F8 was generated targeting the protective F4 fraction from the supernatants, which greatly reduced clinical signs [63]. This mAb was able to completely protect against lethal challenges with *B. divergens* isolates Rouen 1987 (Rouen87) and Weybridge 8843 (W8843) but had no effect against another French isolate from Massif Central (6303E) [67,68]. This classic and time-consuming approach was accelerated using recombinant protein.

Another effective approach involved using recombinant Bd37 protein combined with a hydrophobic peptide, such as saponin, which mimics the merozoite surface and stimulates a strong immune response that inhibits parasite development and disease onset. This formulation was tested successfully in gerbil protection experiments. Bd37 highlights the importance of GPI-anchored proteins in *Babesia* biology and their complex interactions with the host, warranting further investigation [63].

### 5.7. Conclusion

Significant progress has been made in developing vaccines against *Babesia* species, particularly in animals. Whole-parasite vaccines, including attenuated and inactivated forms, have shown promise by presenting a broad range of antigens, but challenges remain regarding safety, production, and immune response consistency. Subunit vaccines targeting specific surface antigens like BmSA1 (*B. microti*) and Bd37 (*B. divergens*) demonstrate potential through their ability to induce targeted immune responses, though efficacy varies and requires further optimization, especially regarding adjuvants and strain variability. Despite advances in animal vaccine research, no vaccine for human babesiosis is yet commercially available. Continued research and development are essential to overcome existing limitations and achieve effective, safe human vaccines in the future.

## 6. Ehrlichia

### 6.1. Introduction

The increasing incidence of human monocytic ehrlichiosis (HME), caused by *Ehrlichia chaffeensis*, highlights the urgent need for effective prevention strategies. Despite growing knowledge about the pathogen’s transmission and pathogenesis, no commercially available vaccines currently exist to protect humans against infection [69,70].

### 6.2. Vaccine Development: Molecular Basis and Immunogenicity

Advances in structural vaccinology have enabled the identification of highly immunogenic surface proteins of *E. chaffeensis*, such as OMP-1 (P28-19) and heat shock protein Hsp60. These proteins elicit strong CD4+ T cell responses and IFN-γ production, indicating their potential as vaccine components. Studies in mice have demonstrated that vaccination with recombinant Hsp60 or P28-19 significantly reduces bacterial burden following infection [69].

In addition to classical surface proteins, the *E. chaffeensis* invasin EtpE has garnered particular attention. An experimental vaccine containing the recombinant C-terminal domain of EtpE (rEtpE-C) effectively induced immune responses in dogs and significantly accelerated pathogen clearance following tick exposure. Vaccinated animals exhibited high antibody titers, increased IFN-γ expression, and robust immunological memory activation [70].

Simultaneously, the use of a live attenuated vaccine (MLAV), based on a strain with a mutation in the phtcp gene, demonstrated effective and long-lasting protection in dogs—up to 12 months post-immunization. This vaccine elicited strong CD4+ T and IgG responses and reduced both the frequency and duration of bacteremia following wild-type challenge [71].

### 6.3. Molecular Immune Responses

Protection against ehrlichiosis primarily relies on a Th1-type immune response involving CD4+ T cells and cytokines such as IFN-γ. Animal models have also shown that IL-17 may play a relevant role in the anti-ehrlichial response, potentially supporting cellular immunity [71,72]. Neutralizing antibodies, especially IgG2 and IgG3 subclasses, are also essential for clearing circulating bacteria and preventing further spread [69].

In one study, immune responses induced by experimental vaccines were compared to those resulting from natural infection. The vaccines generated stronger and more targeted cellular and humoral responses, underscoring the advantage of prophylactic immunization over natural immunity [73].

### 6.4. Public Acceptance and Public Health Strategies

Despite compelling data on vaccine efficacy in animal models, public acceptance of vaccines against ehrlichiosis remains limited. This is partly due to low awareness of ehrlichiosis, which is frequently misdiagnosed or undiagnosed due to its nonspecific clinical presentation [70]. Thus, public health campaigns and education efforts are crucial for raising awareness of tick-borne infections and promoting vaccination—especially in endemic regions.

### 6.5. Challenges and Prospects

The most pressing obstacles in ehrlichiosis vaccine development include the following: the lack of clearly defined immunodominant human epitopes; limited availability of animal models that accurately mimic human infection; and uncertainty about the long-term durability of immune protection [69,70,71].

Conversely, the continued development of structure-based epitope design, subunit vaccines with advanced adjuvants (e.g., ISCOM), and a deeper molecular understanding of *Ehrlichia* pathogenesis open promising new avenues for effective preventive measures [69,70,71].

## 7. Rickettsia

### 7.1. Rickettsia General Information

*Rickettsia* is a genus of obligate intracellular Gram-negative bacteria transmitted by arthropods that feed on the blood of small mammals [74]. *Rickettsiae* are therefore zoonotic pathogens transmitted to humans through the bites of ticks, fleas, and lice [75]. When present in humans, these bacteria cause an infection characterized by fever, rash, headache, myalgia, and an eschar at the site of inoculation by the feeding tick or mite [76]. The disease itself can follow various courses, ranging from mild to even fatal, and often mimics other infections [77,78]. The diagnosis of rickettsioses generally relies on serological methods, with the indirect immunofluorescence assay being the most commonly used technique [79]. Currently, doxycycline is the treatment of choice for infections caused by *Rickettsia*, although azithromycin has also been shown to be an effective alternative, with fewer side effects and added safety for pregnant women and children [79,80].

### 7.2. Rickettsia Classification

These bacteria are classified as *Alphaproteobacteria* and are part of the family *Rickettsiaceae*, which consists of two genera: *Rickettsia* and *Orientia*. *Rickettsia* was further classically divided into typhus and spotted fever groups based on the phenotypic characteristics [81]. Recent phylogenetic analyses of the rOmpA gene and rOmpB gene sequences have suggested a new, more complex classification, suggesting 27 *Rickettsia* species [82]. However, that was later simplified based on core genome alignments that concluded that *Rickettsia* should be classified only into 9 distinct species instead of the previously proposed 27 [83].

### 7.3. Whole-Cell (Killed) Rickettsia Vaccines

The development of vaccines against rickettsial diseases dates to the epidemic of typhus from the early 20th century [84,85]. Early vaccines used whole killed *Rickettsia* bacteria extracted from infected louse or eggs [84,86]. Despite the fact that those vaccines were difficult to produce and had various side effects, they were used by soldiers during World War II (WWII) [87]. Additionally, later studies have shown that they can offer protection against the pathogen in canine animals and produce a cell-mediated immune reaction in humans [88,89,90]. In the years after WWII a purified, formalin-killed *Rickettsia* vaccine was produced in chicken fibroblasts that had fewer side effects in humans compared to the previous vaccine made using yoke sacs [91]. A study conducted by Kenyon et al., in which they tested formalin-inactivated Rickettsia vaccines’ effects on rhesus monkeys, concluded that complete protection was not achieved by a single injection by any of the whole-cell killed vaccines; however, all of the formalin-inactivated vaccines protect from lethal infection [92]. These findings were later confirmed in another animal study performed by Gonder et al., and overall, it was concluded that while the whole-cell Rickettsia vaccine reduces the severity of the disease and protects from life-threatening outcomes, it does not sterilize the infection [93].

### 7.4. Live Attenuated Rickettsia Vaccines

Live attenuated vaccines are generally thought to produce longer, stronger, and more durable immune responses compared to whole-cell killed vaccines due to the fact that they provide stronger CD8+ T cell and TH1-type responses and exhibit stronger mucosal activity [94,95]. During rickettsia isolation and continuous passage through embryonated chicken eggs, a mutation occurred in the *R. prowazekii* strain Madrid E that caused a change in methyltransferase, which functions to methylate the surface proteins, including OmpB [96,97]. This vaccine provided long-lasting immunity in humans for up to 5 years. Other studies suggest that this methyltransferase can revert to wild type following animal transmission [96,98,99]. Since the 2000s, no live or killed rickettsia vaccine has been in use in humans [87,88].

Modern live attenuated *Rickettsia* vaccines are created by using genetic tools to upregulate virulence genes [88,100]. In a study conducted by Arroyave et al., they created a *Rickettsia parkeri* mutant (strain 3A2). They showed that a single dose administration of this strain resulted in protection from fatal murine spotted fever rickettsiosis in mice [101]. Driskell et al. conducted another study wherein they targeted a deletion of the phospholipase D (pld) gene in *R. prowazekii,* which created an altered form that is not pathogenic in guinea pigs and produces a protective immunity [100]. Another study performed by Olsen et al. showed that a single dose of live attenuated *Rickettsia* vaccine produced a sufficient immune response in Atlantic salmon depending on the water temperature during immunization [102]. To sum up, live attenuated *Rickettsia* vaccines are currently not used in humans; however, following the success in gene knockout methods in animals, similar approaches can lead to the development of a new live attenuated vaccine in humans [103].

### 7.5. Subunit and Recombinant Protein Vaccines and Rickettsia Vaccines

Rickettsial subunit vaccines have focused on autotransporter proteins that are members of the surface cell antigen, most notably OmpA and OmpB, which are clearly recognized by antibodies in animals and humans [104,105]. In another study, Summer et al. demonstrated that OmpA vaccines protect guinea pigs against two types of *Rickettsia* [106]. In another study, Jiao et al. used a recombinant OmpA fragment vaccine and showed that it induced strong antibody and T cell responses in mice and guinea pigs [107]. OmpB from *R. conorii* vaccines have also been shown to be efficacious in preventing fatal *R. rickettsii* infection in mice [108,109]. However, Cardwell et al. performed a study in which the vaccine consisting of OmpB from *R. conorii* generated high-titer antibodies but did not protect mice from *R. rickettsii*. Therefore, the conclusion was made that nearly identical OmpB sequences are not sufficient for protection [108].

Other rickettsial surface proteins have shown promise in creating a vaccine. These include the Adr1/Adr2, YbgF, TolC, and GroEL. Gong et al. showed that recombinant *R. ricketsii* Adr2 significantly lowers rickettsial presence in spleen, liver, and lungs of mice due to a Th1 response [110]. In another study performed by Gong et al., the researchers demonstrated that recombinant protein YbgF from *R. heilongjiangensis* resulted in IFNγ release by both CD4+ and CD8+ T cells, prolonged immunoglobulin production, and overall reduced bacterial load in mice [111]. Gong et al. also demonstrated that immunization with rAdr1, rOmpW, or rPorin 4 significantly reduced rickettsial load in mice [112].

### 7.6. DNA and mRNA Rickettsia Vaccines

Rickettsial DNA and mRNA vaccines work by delivering genetic templates that encode rickettsial antigens into the host cells, which later leads to the expression of rickettsial antigens in situ [113]. These types of vaccines have been shown to be efficacious in animals [114]. In a study conducted by Crocuet-Valdes et al., it was shown that a recombinant DNA vaccine protected against lethal rickettsial infection in mice [115]. Additionally, in cultures of human cells, rickettsial DNA vaccine stimulated IFN-γ+ CD4 and CD8 lymphocytes [116]. However, DNA vaccines are currently only used in experimental models. No Phase I or II trials have been reported, so they remain in the preclinical phase. No mRNA vaccine has yet been developed. Nevertheless, given their ability to combine multiple antigenic determinants from different proteins, mRNA vaccines could be a promising future strategy for protecting against rickettsial diseases [117].

### 7.7. Viral Vector and Nanoparticle Rickettsia Vaccines

Viral vector vaccines are commonly used in vaccinology; however, currently, this technique has not been applied in the case of *Rickettsia*. Viral vector rickettsial vaccines could potentially use human or other members of the Hominidae family adenovirus to deliver the *Rickettsia*’s surface proteins [87].

Nanoparticle vaccines are a novel approach to vaccinology [118]. Currently, no such vaccine against *Rickettsia* has been developed. However, Ha et al. showed that zinc oxide nanoparticles can be combined with recombinant *O. tsutsugamushi* ScaA, which is a member of the Rickettsiaceae family. The administration of this novel vaccine resulted in antibody responses and protection against lethal infection in mice. This study highlights the potential that nanoparticles have for the future development of new vaccines [119].

## 8. *Anaplasma phagocytophilum*

*Anaplasma phagocytophilum* is a Gram-negative, obligate intracellular alpha-proteobacterium. This cosmopolitan parasite can be transmitted to both animals and humans alike via an *Ixodes* tick bite or blood transfusion [120,121,122]. It is responsible for tick-borne fever (TBF) in sheep and cattle, among others, as well as human granulocytic anaplasmosis (HGA) [123]. Once in the body, the parasites first evade the immune system, and then alter it upon infecting cells like neutrophils, monocytes, and eosinophils, leading to subversion of their bactericidal activity [121,123]. During the early stage, immunosuppressed patients usually experience fever, chills, headache, muscle pain, soreness, and gastrointestinal tract symptoms. In the later stages, the symptoms may progress into bleeding, respiratory distress, organ failure, meningoencephalitis, and finally death, although, with the current available antibiotic treatment, HGA mortality rate ranges between 0.5 and 1% [121,122,124]. Animals with TBF are also shown to have greater susceptibility to secondary infections, but other symptoms include abortions and severe reduction in milk production in dairy cattle, which can lead to great economic losses in the livestock industry across many countries [123,125]. As for now, there is no vaccination for *Anaplasma phagocytophilum*, although numerous trials have been performed, some of which have shown promising results.

An attempt at subcutaneous immunization with inactivated *A. phagocytophilum* indicated that, although lambs used in the trial showed anamnestic response, it was too meager to grant them protective immunity against TBF. The authors of the paper concluded that the method they used in the trial exposed the immune system of the test subjects mainly to the dominant antigens of the pathogen, thus failing to elicit a sufficient immune response. Hence, a successful vaccine should be directed at shared or subdominant antigens conserved amongst all strains of *A. phagocytophilum* [126]. Thus, some research puts heavy emphasis on major surface protein 4 (MSP4) and heat shock protein 70 (HSP70)—two highly conserved proteins localized on the bacterial membrane, which participate in tick–pathogen and host–pathogen interactions. Lambs immunized with *A. phagocytophilium* MSP4 and MSP4-HSP70 recombinant proteins had similar results to those reported in the lambs immunized with inactivated *A. phagocytophilum* [127]. However, this MSP4 recombinant vaccine yielded a protective response in rabbits. It is likely that even with MSP4 antigens in the recombinant vaccine, the lambs were still directing an IgG response to non-protective epitopes. Perhaps additional studies focused on developing chimeric antigens using quantum vaccinomics are needed. Upon testing recombinant chimeric antigens, which contained candidate protective epitopes identified in rabbits, IgGs from sheep were shown to affect *A. phagocytophilum* infection. This supported the idea of further future testing to design new chimeras with host-specific adjustments, which would account for host-related factors, including age, sex, genetic factors, microbiota, pregnancy, and immune history [128].

Similar findings were observed with immunization using *A. phagocytophilum* surface protein (Asp14), which further affirmed that both the test subject and the method can give varying results. One study has shown vaccination with recombinant Asp14 to be ineffective in preventing and reducing the effects of TBF infection in sheep. However, another study achieved promising results in mice immunized with a cocktail of keyhole limpet hemocyanin-conjugated peptides corresponding to the AipA, Asp14, and OmpA binding domains in alum [129,130]. Although it was proven that OmpA59-74 did not contribute to the milder outcome of the *A. phagocytophilum* infection, antibodies specific for Asp14113-124 and AipA9-21 were observed to reduce the pathogen burden in mice, seen with Asp14113-124 more so than AipA9-21 [130].

Other researchers have focused on three conserved membrane proteins subdominant in *Anaplasma* species. VirB9-1, VirB9-2, and VirB10—part of the Type 4 secretion system (T4SS) that is crucial for invasion and survival in host cells for many intracellular bacterial species. These gene products were introduced to murine test subjects in the form of intramuscular vaccination with plasmid DNA. The VirB10 antigen proved to be partially protective due to the induction of interferon–gamma response, suggesting it to be a potential vaccine candidate for both *Anaplasma phagocytophilum* as well as other intracellular pathogens utilizing T4SS [131].

A different approach was presented in a study attempting to utilize *I. scapularis* Organic Anion Transporting Polypeptide 4056 (IsOATP4056). The approach taken here was in the form of an anti-vector vaccine to prevent parasite transmission. The above-mentioned arthropod membrane transporter was shown to be upregulated in salivary glands of ticks infected with *A. phagocytophilum* and during the transmission of the pathogen from infected ticks to vertebrate hosts. The study used murine test subjects to check the efficacy of EL-2 and EL-6 antibodies against the bacteria and found that high doses of EL-6, an antibody targeting the largest extracellular loop of IsOATP4056, impaired the transmission of *A. phagocytophilum* from infected ticks to mice. Moreover, there was evidence that vectors were affected as well. The results showed that ticks that fed on the immunized test subjects experienced reduced molting efficiency and that EL-6 antibody had a cytotoxic effect on tick cells infected by the pathogen, reducing their bacterial loads. Furthermore, the same cytotoxicity has not been found in human keratinocytes, suggesting that EL-6 antibody immunization could be safe for humans, making it a strong candidate for further development of an anti-vector vaccine [132].

Research into vaccines against *Ehrlichia*, *Rickettsia*, and *Anaplasma* remains limited due to challenges such as a lack of well-characterized animal models, antigenic variability, and generally low prioritization compared to Lyme disease and TBE.

## 9. Discussion

In our review, we primarily focused on the vaccine development process for different vaccine types, as shown in Table 3, and the associated challenges with it. However, developing a vaccine does not necessarily lead to success in terms of providing global protection against tick-borne diseases. Many other factors come into play, such as implementation strategies, public attitudes towards mass vaccination, market analysis, and financial viability. For each vaccine, calculations must be made to determine whether it is financially beneficial for the state to vaccinate the entire population or only selected high-risk groups, and whether the vaccine should be free or chargeable to patients. Such estimates can be made in various ways; one method is the Markov model. This model has already been used to assess the cost-effectiveness of tick-borne encephalitis vaccination. Studies conducted in Slovenia showed that vaccinating adults aged 18–80 (including booster doses) is cost-effective from the perspective of the healthcare payer. However, Swedish studies using the Markov model did not demonstrate cost-effectiveness for all age groups. It should be noted that financial models assume a minimum number of people will be vaccinated. Considerable financial outlay is required for vaccine production and clinical and preclinical trials [133]. If there is insufficient public interest in the vaccine, work on it may be suspended. This was the case with the LYMErix^®^ vaccine. Low public acceptance and concerns led to commercial failure, prompting the manufacturer to withdraw the vaccine from the market [134]. D. Lewandowski et al. conducted a study to identify the key factors influencing people’s willingness to accept a future Lyme disease vaccine. The study revealed that the general attitude towards vaccines and the perception of the risk of contracting Lyme disease are the main factors influencing the decision to vaccinate. Low public awareness of tick-borne diseases is also significant [135]. Education and future opportunities may be helpful in this regard. Combining vaccines, e.g., against Lyme disease and tick-borne encephalitis (TBE), into one means fewer injections and is potentially an attractive solution for patients. There is also hope in the development of new technologies that will be cheaper to produce and less controversial due to side effects. However, all this requires considerable research and years of work.

## 10. Conclusions

This review summarizes information on vaccines for the following diseases: tick-borne encephalitis, Lyme disease, *Babesia*, *Ehrlichia*, *Rickettsia*, and *Anaplasma*. Currently, only a vaccine for tick-borne encephalitis is available: FSME-IMMUN^®^ and Encepur^®^ contain inactivated TBE virus. Research is also underway on new inactivation methods and vaccines based on virus-like particles (VLPs). A VLA15 vaccine targeting the OspA protein for use against Lyme disease is in Phase III clinical trials. Less advanced, but ongoing, research is being conducted on OspC and CspZ antigens. Currently, there are no vaccines against other tick-borne diseases. In preclinical studies, attenuated and inactivated vaccines for *Babesia*, as well as subunit vaccines, are being tested. For *Ehrlichia*, the OMP-1 (P28-19), Hsp60, and EtpE antigens are being tested. Attempts have been made to create live attenuated and subunit vaccines (OmpA, OmpB), and DNA/mRNA vaccines against *Rickettsia.* The antigens being tested against *Anaplasma* are MSP4, HSP70, and Asp14.

The steadily increasing impact of tick-borne diseases on human and animal health and the economic factors on healthcare and agricultural productivity provide strong motivation for a much-increased investment into research and development of a number of vaccines. The current technical and financial challenges to create next-generation, efficacious products are well-recognized. Despite those challenges, steady advances in systems biology understanding of host–pathogen interactions, along with bioinformatic-assisted approaches to vaccine development, can potentially overcome these challenges eventually.

## Figures and Tables

**Table 1 vaccines-13-00990-t001:** Landscape of experimental and licensed Lyme disease vaccines, including their antigen targets, formulation, clinical status, key efficacy/safety findings, and notable development considerations.

Vaccine/Strategy	Antigen(s) and Platform	Formulation/Dose and Schedule Tested	Clinical Phase/Status	Key Notes	Safety/Reactogenicity	Reference
VLA15	[Table 2] includes					
Monovalent OspA	LYMErix	Dose: 30 µg Schedule: 0, 1, and 12 months	Phase III completed; FDA approved 1998 → Withdrawn in 2002 due to public perception, not safety concerns	N = 10,936 (15–70 y), double-blind, placebo-controlled, randomized trial Outcome: 49% efficacy after 2 doses; 76%	Mild local/systemic reactions; no serious adverse events attributed to the vaccine	[34]
		Dose: 30 µg Schedule: 0, 1, and 12 months		N = 956 (17–72 y), open-label, randomized trial Outcome: 2.8-fold increase in GMT; 90% ≥ 1400 EL.U/mL, 99% ≥ 400 EL.U/mL; predicted seasonal protection	Mild local/systemic reactions; no serious adverse events attributed to the vaccine	[45]
	ImuLyme^TM^	Dose: 30 µg Schedule: 0, 1, and 12 months	Phase III completed; never marketed due to market failure of LYMErix	N = 10,305 (21–79 y), double-blind, placebo-controlled randomized trial Outcome: 68% efficacy (2 doses); 92% efficacy (3 doses); well tolerated	Mostly mild local AEs	[35]
		Dose: 30 µg Schedule: 0, 1, and 12 months		N = 1634 (18–94 y), double-blind, placebo-controlled randomized trial Outcome: 40% efficacy (Year 1); 37% (Year 2); efficacy limited to <60 y (50%)	Mostly mild local/systemic AEs	[26]
Multivalent OspA	6 serotypes	3 × 10 µg IM injections at days 0, 28, 56; Alum adjuvanted	Preclinical (Mice)	High IgG to all 6 ST; lower redness/fever vs. monovalent vaccines	Mild reactogenicity; no serious issues	[37]
	Multivalent OspA vaccine: Three recombinant lipidated fusion proteins	Dose: 30, 60, and 90 µg Schedule: 0, 1, 2, and 12 months	Phase I/II trial; development halted	N = 300 (18–70 y), double-blind, randomized, dose-escalation trial Outcome: 30 μg adjuvanted = best tolerated; no SAE; post-booster IgG GMT 26,143–42,381 EL.U/mL	Well tolerated; local injection site pain common; no vaccine-related serious adverse events	[46]
		Dose: 30 and 60 µg Schedule: 0, 1, 2, and 6 0, 1, 2, and 9–12		N = 350 (18–70 y), double-blind, randomized, dose-escalation trial Outcome: GMT post-booster ranged from 26,143 (95% CI: 18,906–36,151) to 42,381 (95% CI: 31,288–57,407)	Well tolerated; local injection site pain common; no vaccine-related serious adverse events	[47]
OspC	Chimeritope Tetravalent OspC	In vitro bactericidal recombinant proteins, 25 µg IP × 3 doses at days 0, 21, 42	Preclinical	Mice (C3H/HeN); broad anti-OspC bactericidal IgG; bactericidal activity; broad protective effect; antigenic diversity limits stand-alone use	n/r (animal study)	[27]
CspZ	CspZ, VLP-CspZ, VLP-CspZY207A/Y211A	Modified VLP-CspZ + Titer Max Gold 25 µg IP × 3 doses at days 0, 14, 28	Preclinical	Mice (C3H/HeN); modified VLP-CspZ; superior immunogenicity compared to unmodified VLP-CspZ; significantly higher borreliacidal antibody titres (50% killing at 1:395 serum dilution vs. 1:143); active immunization and passive antibody transfer effectively combat *Borrelia* infection; enhanced efficacy; elimination of factor H-binding activity	n/r (animal study)	[28]
FtlA/FtlB lipoproteins	FtlA and FtlB (membrane-associated PF12 lipoproteins)	Recombinant proteins; rat hyper-immune sera	Preclinical	Dogs, horses; Anti-FtlA/B serum ≥ 65% borreliacidal; antibodies detected in infected dogs/horses; anti-Ftl antibodies detected in 61–84% of infected dogs and horses; levels increased up to 497 days, indicating persistent in vivo expression	n/r (animal study)	[48]
Reservoir-targeted OspA RTV	rOspA-based oral vaccine	Environmental deployment; field study	Operational field trial	Wild white-footed mice-*Peromyscus leucopus*; reduction (76%) in *B. burgdorferi* infection in nymphal ticks after 5 years; increased anti-OspA seroprevalence in vaccinated mice (mean OD450: 0.391 vs. 0.229; *p* = 0.002); protective titre (OD450 ≥ 1.0) reached 21–33% in vaccine plots vs. ~5% in controls	n/r (animal study)	[30]
P35/P37	P35 and P37 (in vivo-expressed proteins of *B. burgdorferi*)	Passive transfer of sera containing antibodies to P35 and P37	Preclinical	Mice; P35 and P37; inducing strong IgG responses(in vivo); promising for both vaccine and diagnostic development; antisera provided protective immunity in mice both prophylactically and therapeutically (post-infection); seroconversion in 100% of infected mice	n/r (animal study)	[33]
BBA52	BBA52 (outer membrane protein of *Borrelia burgdorferi*)	Surface-exposed outer membrane protein BBA52; native protein with disulfide-linked oligomeric structure	Preclinical	Mice (C3H/HeN); vector–host transition; active immunization: mice immunized with BBA52 protein; passive immunisation: transfer of antibodies into ticks; antibodies showed no activity (in vitro); passive immunization as well	n/r (animal study)	[43]

(Abbreviations: OspA/OspC, outer surface protein A/C; ST, OspA serotype; FH, factor H; KLH, keyhole limpet hemocyanin; IP, intraperitoneal; n/r, not reported; GMT, geometric mean titer; SAE, serious adverse event; TBE, tick-borne encephalitis; QALY, quality-adjusted life year).

**Table 2 vaccines-13-00990-t002:** Clinical development of the multivalent OspA vaccine VLA15.

Stage	Key Findings	Reference
Preclinical (murine ± guinea-pig)	▸ Three-dose priming protected mice against *B. burgdorferi* (ST1), *B. afzelii* (ST2), *B. bavariensis* (ST4), and *B. garinii* (ST5, ST6). ▸ Growth inhibition assay demonstrated functional activity against all six OspA serotypes (ST1-6). ▸ A booster given five months after priming produced higher peak IgG/SBA titres and longer serum half-lives.	[38]
Phase I (179 adults, 18–39 y)	▸ Three monthly doses (0–1–2 mo; 12 µg–90 µg) were safe and well tolerated. ▸ Broad, serotype-wide IgG generated, but titres declined to near baseline by month 11. ▸ A month-13 booster elicited a strong anamnestic surge that again waned within ~6 mo.	[51]
Phase II—dose and schedule optimization	▸ Two randomized multicentre trials (USA + EU) compared 135 µg vs. 180 µg and 0–1–2 mo vs. 0–2–6 mo schedules (total n ≈ 821). ▸ Total of 180 µg on a 0–2–6 mo schedule produced the highest GMT and the slowest decay; 0–1–2 mo was less durable. ▸ Reactogenicity was chiefly mild local pain/tenderness; systemic events were comparable with placebo.	[51]
Phase II—booster extension	▸ Total of 59 participants (received VLA15 (n = 39) or placebo (n = 19)) primed with 180 µg (0–2–6 mo) were re-randomized 12 mo later (month 18) to receive booster or placebo. ▸ Booster raised GMT **≈** 3–4-fold relative to the previous peak and remained significantly above placebo at month 30. ▸ Safety profile unchanged; no vaccine-related serious adverse events.	[52]
Phase III (ongoing)	▸ Pivotal VALOR efficacy trial (NCT05477524) enrolling individuals ≥ 5 years in Europe and the USA. ▸ Double-blind, placebo-controlled design; regimen = 180 µg at months 0, 2, 5–9 with a booster ~12 mo later. ▸ End-points: efficacy, safety, immunogenicity, and lot consistency; study currently in progress.	[53]

(*Abbreviations*: GMT, geometric mean titer; SBA, serum bactericidal assay; ST, OspA serotype).

**Table 3 vaccines-13-00990-t003:** Comparison of different types of vaccines.

Vaccine Platform	How Does It Work?	Advantages	Disadvantages	Reference
Subunit (Protein/Antigen)	Pathogens’ immunogenic proteins are produced using conventional biomedical methods. They are also produced using recombinant DNA technology.	▸ Can be used in patients with compromised immunity. ▸ With a reduced likelihood of adverse reactions, it is considered relatively safe. ▸ A high neutralizing-antibody titer compared to inactivated virus vaccines.	▸ They are less immunogenic than live attenuated vaccines and require an adjuvant to stimulate an immune response. ▸Long-lived immunity requires multiple doses.	[136]
Inactivated (Killed)	Genetic material from pathogens is destroyed (through heat, radiation, or chemically, e.g., formalin and phenol). Pathogens cannot reproduce.	▸ Requires less severe storage conditions. ▸Compared to attenuated vaccines, this is safe for use in patients with compromised immune systems.	▸ Large amounts of antigen and booster doses are required to achieve the desired immunity. ▸They are less immunogenic than live attenuated vaccines, and they are not very effective at stimulating a cellular response. They also require an adjuvant to boost the immune system.	[136]
Vector-based	Viruses that have been altered and are not related to each other, and which encode one or more antigens.	▸ It presents the desired antigens in their natural form to the immune system. ▸ It has a better safety profile than many live attenuated virus vaccines and is more immunogenic than inactivated virus vaccines.	▸ There is a risk of host–genome integration. ▸ The effectiveness of the vaccine can be reduced by previous immunity to the vector due to exposure to the virus and the production of neutralizing antibodies.	[136]
mRNA	Contain genetic materials (mRNA) from the target pathogen.	▸ Host cells produce antigens. ▸ There is less likelihood of biological changes occurring during production in the vaccinated host. ▸The mRNA vaccines can trigger strong responses from both T-helper 1 (Th1) cells and B cells.	▸ Myocarditis is one of the possible complications that can arise from mRNA vaccinations. ▸ It is hard to distribute RNA molecules worldwide because they are unstable at high temperatures.	[136]
Live Attenuated	The whole viable pathogen. It has reduced virulence.	▸ This is the closest mimic of a natural infection. It is a good teacher for the immune system. ▸Immunization can be achieved with one or a few doses, without the need for an adjuvant. ▸It usually provides lifelong immunity and a strong induction of both cellular and humoral immunity.	▸ The potential for virulence reversion resulting from back mutations. ▸Administration of this product is not recommended for patients with compromised immune systems. ▸ Critical storage conditions are required to maintain potency.	[136]
Virus-Like Particles (VLPs)	Recombinant proteins that are structurally very similar to natural virions.	▸ A wide range of production systems can be used to produce VLPs, making them flexible in terms of production conditions. ▸VLPs are not infectious because they contain no viral genomes.	▸VLP vaccines contain many proteins and the degree to which they induce an immune response is unclear.	[136]

## Data Availability

Not applicable.

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
