# Peer review of "What Stage Are We at in the Development of Vaccines Against Tick-Borne Diseases?"

_vaccines, 2025, doi:10.3390/vaccines13090990_

Round 1

Reviewer 1 Report

Comments and Suggestions for Authors Add a comparative table of technology platforms (e.g., inactivated, VLP, mRNA) that describes their respective strengths and limitations.   Provide a discussion of public health implementation strategies.   Review and align the editorial style for consistency and clarity. Include a section addressing "Cross-Cutting Challenges and Future Opportunities."

Comments on the Quality of English Language The manuscript is generally understandable, and the English is adequate to convey the authors' ideas. However, there are several grammatical errors and awkward phrases throughout the text that occasionally disrupt the flow. A thorough linguistic review, preferably conducted by a native English speaker or a professional editor, would improve clarity and readability before publication.

Author Response

Thank you for taking the time to assess our manuscript! We've addressed all the concerns you raised.

Comments 1: Review and align the editorial style for consistency and clarity.
Response 1: Thank you for your comment. The style of the entire Table 1 has been improved. The text in each cell has been centred. We also noticed that some of the text in this table was justified and some was not. We have removed the justification. The style of Tables 2 and 3 has been improved in the same way. We are also happy to change other things and will take your suggestions on board. Unfortunately, we have no idea what else we could change in terms of style. If you still feel it is necessary, please provide more details. We have also changed a few sentences. I will mention some of them as examples. 

One relatively recent tick-borne illness is caused by Borrelia miyamotoi… -->
A recently recognized tick-borne illness is caused by Borrelia miyamotoi

This illness should be recognized as a distinct infectious disease… -->
This illness is increasingly recognized as a distinct infectious disease…

Educational interventions, especially in schools, are also important. -->
Educational interventions, particularly in schools, play a critical role in prevention.

We have also changed the titles. All words in the titles now begin with a capital letter.

Comments 2: Add a comparative table of technological platforms (e.g., inactivated, VLPs, mRNA) outlining their respective strengths and limitations.
Response 2: We agree with this comment. Thank you for your suggestion. The table has been included on pages 23 and 24.

Comments 3: Provide a discussion on public health implementation strategies. Include a section addressing "Cross-cutting Challenges and Future Opportunities".
Response 3: Thank you very much for this suggestion. Due to the fact that our work is lengthy, we did not develop the discussion to such an extent. For this reason, we did not include a separate section entitled ‘Cross-cutting Challenges and Future Opportunities’. Nevertheless, in line with your comment, we have written a discussion that was not previously included in the publication. We hope that we have addressed the most important points and that it is sufficient. The discussion can be found on page 25.

Comments 4: The English could be improved to more clearly express the research.
Response 4: The publication was read again. We made many corrections, e.g., we corrected typos and added subjects in several sentences. Another reviewer pointed out all the errors to us. After correcting them, we believe that the work is much better in terms of language. I am attaching the comments of another reviewer in a PDF file to illustrate what has been changed.

Reviewer 2 Report

Comments and Suggestions for Authors

The manuscript provides a comprehensive and timely review of current vaccine development for several major tick-borne diseases (TBE, Lyme disease, Babesia, Ehrlichia, Rickettsia). It is well-structured, with a clear thematic organization and extensive literature coverage. The authors integrate historical perspectives with recent advances, highlighting both successes and remaining challenges. This broad scope will interest researchers, public health practitioners, and vaccine developers.

However, the manuscript would benefit from editorial refinement, consistent formatting, and improved clarity in certain sections. There are also areas where citations are missing or species names need to be italicized. Some sentences are incomplete or lack a clear subject.

The majors strengths of the manuscript are the followings:

-Comprehensive coverage across multiple pathogens, from viral to bacterial and parasitic agents.

- Balanced discussion of historical vaccine efforts and current pipeline candidates, particularly for Lyme disease.

- Integration of immunological insights with practical considerations such as public acceptance and cost-effectiveness.

- Use of recent primary literature (2022–2024) ensures relevance.

- Inclusion of veterinary and reservoir-targeted vaccine concepts, broadening the perspective beyond human immunization.

There are however some weaknesses that need to be improved

Clarity and Language:

- Several sentences are incomplete or lack a clear subject (e.g., TBE section lines 106–108; Lyme section lines 202-203; lines 300-301; ines 664: Rickettsia general info??). These should be rewritten for grammatical completeness; l

- Occasional redundancy between introduction and disease-specific sections could be reduced.

B. Formatting and Scientific Conventions

- All genus and species names (e.g., Ixodes ricinus, Borrelia burgdorferi, Babesia microti) should be italicized consistently.

- Ensure consistent abbreviation formatting upon first use (e.g., TBEV, OspA, CspZ, WWII...).

C. Referencing

-Missing citations are noted in several places (e.g., introduction lines 27–28 regarding landscaping practices; lines 30-31: owning pets and cats.. why not dogs?)

-Ensure that all factual statements, particularly epidemiological data, are referenced.

  • The indication of pages is lacking in reference n°5

D. Scientific Content

- In the Lyme vaccine section, the discussion on correlates of protection is strong, but could further emphasize the public health implications of waning immunity and booster schedules.

- The Babesia section would benefit from a short comparative table summarizing antigens, platforms, and trial outcomes.

- In Ehrlichia and Rickettsia sections, the depth of coverage is less than for Lyme/TBE—consider either expanding these or noting explicitly that research is limited.

Minor Points

- Add space after certain words where currently missing (noted at lines 132–133).

- Remove unnecessary parentheses in lines 251–254, 253–254 as per track-change comments.

- Check for consistency in the use of “phase” when referring to clinical trials (Phase I, Phase II, etc.).

Author Response

Thank you for taking the time to assess our manuscript! We've addressed all the concerns you raised.

Comments 1: Several sentences are incomplete or lack a clear subject (e.g., TBE section lines 106–108; Lyme section lines 202-203; lines 300-301; lines 664: Rickettsia general info??). These should be rewritten for grammatical completeness.
Response 1: Thank you for pointing this out. The sentences have been corrected.

Comments 2: Occasional redundancy between the introduction and disease-specific sections could be reduced.
Response 2: Thank you very much for your comment. We have reread the text and did not notice any repetitions that would bother us. Please specify what should be removed, if necessary.

Comments 3: All genus and species names (e.g., Ixodes ricinusBorrelia burgdorferiBabesia microti) should be italicized consistently.
Response 3: We have made changes throughout the text.

Comments 4: Ensure consistent abbreviation formatting upon first use (e.g., TBEV, OspA, CspZ, WWII...).
Response 4: We ensured consistent formatting, e.g. we removed the space between Osp A -> OspA. We removed the ‘s’ from the abbreviations GMTs -> GMT, SAEs -> SAE.

Comments 5: Missing citations are noted in several places (e.g., introduction lines 27–28 regarding landscaping practices; lines 30-31: owning pets and cats.. why not dogs?).
Response 5: The entire passage up to verse 32 is based on article number 1. With regard to the comment on why not dogs, the sentence has been slightly altered. 

Owning pets also increases the risk of tick bites. -->
More broadly, owning pets also increases the risk of tick bites. 

Comments 6: Ensure that all factual statements, particularly epidemiological data, are referenced.
Response 6: When an entire subsection of the publication was based on a single article, but the text was divided into two paragraphs, we added a number with a citation at the end of the first paragraph.

Comments 7: The indication of pages is lacking in reference n°5.
Response 7: This article does not have the usual page numbering, so it cannot be added to the bibliography. Instead, we have added doi.  2017;83(15):e00994-17. doi:10.1128/AEM.00994-17

Comments 8: In the Lyme vaccine section, the discussion on correlates of protection is strong, but could further emphasize the public health implications of waning immunity and booster schedules.
Response 8: Thank you for your suggestion. We have added the following passage: From a public health perspective, waning antibody levels mean that booster doses will likely be required to sustain protection over time. Clear guidance on booster schedules will therefore be critical not only for maintaining individual immunity, but also for ensuring confidence in large-scale vaccination programs

Comments 9: The Babesia section would benefit from a short comparative table summarizing antigens, platforms, and trial outcomes.
Response 9: Thank you very much for your suggestion. However, following the recommendation of another reviewer, we have added a table on pages 23 and 24. Therefore, we decided not to create a fourth table so that they would not dominate the text.

Comments 10: In Ehrlichia and Rickettsia sections, the depth of coverage is less than for Lyme/TBE—consider either expanding these or noting explicitly that research is limited.
Response 10: Thank you very much for your suggestion. We have added an explanatory sentence at the bottom of page 22:

Research into vaccines against Ehrlichia, Rickettsia and Anaplasma remains limited due to challenges such as lack of well-characterized animal models, antigenic variability, and generally low prioritization compared to Lyme disease and TBE.

Comments 11: 

- Add space after certain words where currently missing (noted at lines 132–133).

- Remove unnecessary parentheses in lines 251–254, 253–254 as per track-change comments.

- Check for consistency in the use of “phase” when referring to clinical trials (Phase I, Phase II, etc.).
Response 11: We have corrected everything according to your instructions.

Reviewer 3 Report

Comments and Suggestions for Authors

Please see the attached file with suggestions on how to improve this manuscript.  I am very eager to see this in print, since it is a timely review and will contribute greatly to alerting the scientific community of the overarching issues and successes in this expanding area of public health concern.  

Comments on the Quality of English Language

The authors will see in the attached comments that there are areas where the quality of English is acceptable and other areas that need improvement.

Author Response

Thank you for taking the time to assess our manuscript! We've addressed all the concerns you raised. We are very pleased to read such a review. We feel that our work has been appreciated.

Comments 1: Does the Journal not have a policy to italicize scientific names of genus and species (e.g. Ixodes scapularis). If this is the stylistic policy, then all of the Linnean names need to be treated this way.
Response 1: All names have been written in italics as recommended.

Comments 2: There were many instances of combined words, or non-hyphenated words, and typographic errors that needed to be addressed. When citing papers in the text, it is not necessary to use the first names of the first author in the citation. 
In some of these sections, the English grammar and syntax are fine, and in other sections, there is a notable difference in that quality. When these are combined with the issues related to misspellings, etc., it distracts from the reader’s flow in focusing on the real information.
Response 2: Thank you very much for such a detailed review. Thank you for taking the time to note down the errors along with the line numbers. Due to the number of errors, I am not listing all the corrections here, but they have been made exactly as in the review.

Comments 3: Summary — I would have liked to have seen some larger overall analysis of the state of these vaccine efforts, as in what are chances of success, what are the prime impediments, what might be the cost or market forces that help or hinder the advancement and if none of these are successful, for whatever reasons, where does that leave the state of how we
achieve protection or reduction of the impact of tick-borne diseases on society (humans and animals). This would serve as a better call to action on needs, future directions, etc.
Response 3: We found this suggestion very interesting. It would certainly be a very good addition to the publication. Unfortunately, however, we are unable to describe the chances of success, as this would require professional market analysis and business knowledge. However, to complement our article at least a little, we have added a discussion at the end on page 25.

Round 2

Reviewer 3 Report

Comments and Suggestions for Authors

Thank you once again for this valuable summary of current efforts in vaccinology for tick-borne diseases.  It will add greatly to the literature.  I note a few typographical and word usage issues.  I also offer some suggestions on minor improvements on readability.  Finally, I still suggest that there could be a better summary statement and provide one example below.  The summary section does a good job in collecting the specific outcomes of each disease, but a positive statement for what lies ahead would be beneficial.

Line 448 – change to:  “six-serotype VLA15 panel”

Line 771 – change to:  “protects from..”

Line 726 – change to: “they created..”

Line 729 – change to: “phospholipase”

Line 734 – change to: “To sum up, live attenuated..”

Line 749 – change to “from R. rickettsii..”

Line 753 – remove hyphen  “- presence in spleen”

Line 756 change to : “CD8+ T cells, prolonged…”

Line 766 – change to: “..against lethal rickettsial infection in mice.  Additionally..”

Line 769 – change to: “..trials”

Line 772 – change to:  “protein, they could be..”

Line 796 – this is a run on sentence.  I suggest splitting into two sentences after discussing the gastrointestinal symptoms in the early stages and then saying “In the later stages….”

Line 818 – the language suggests that the subunit vaccine was still as unprotective as the inactivated vaccine for A. phagocytophilium.   Is that correct?  The language is a bit vague.  I would change the next sentence about the outcome in rabbits to say:  “However, this MSP4 recombinant vaccine yielded a protective response in rabbits.  It is likely that even with MSP4 antigens in the recombinant vaccine, the lambs were still directing IgG response to non-protective epitopes.  Perhaps additional studies focused on developing chimeric antigens using quantum vaccinomics is needed.”

Line 842 – change to: “that is crucial…”

Line 844 – change to “The VirB10 antigen…”

I had suggested previously that in the summary statement the authors provide some overarching comment on the need to continue strong efforts for vaccines against these diseases.  As an example, I would offer:

“The steadily increasing impact of tick-borne diseases on human and animal health and the economic factors on healthcare and agricultural productivity provide strong motivation for a much increased investment into research and development of a number of vaccines.  The current technical and financial challenges to create next-generation, efficacious products are well-recognized.  Despite some progress as discussed here, steady advances in systems biology understanding of host-pathogen interactions along with bioinformatic-assisted approaches to vaccine development can potentially overcome these challenges.”

Author Response

Thank you once again for your time and detailed suggestions.

Comments 1:
Line 448 – change to:  “six-serotype VLA15 panel”
Line 771 – change to:  “protects from..”
Line 726 – change to: “they created..”
Line 729 – change to: “phospholipase”
Line 734 – change to: “To sum up, live attenuated..”
Line 749 – change to “from R. rickettsii..”
Line 753 – remove hyphen  “- presence in spleen”
Line 756 change to : “CD8+ T cells, prolonged…”
Line 766 – change to: “..against lethal rickettsial infection in mice.  Additionally..”
Line 769 – change to: “..trials”
Line 772 – change to:  “protein, they could be..”
Line 796 – this is a run on sentence.  I suggest splitting into two sentences after discussing the gastrointestinal symptoms in the early stages and then saying “In the later stages….”
Line 818 – the language suggests that the subunit vaccine was still as unprotective as the inactivated vaccine for A. phagocytophilium.   Is that correct?  The language is a bit vague.  I would change the next sentence about the outcome in rabbits to say:  “However, this MSP4 recombinant vaccine yielded a protective response in rabbits.  It is likely that even with MSP4 antigens in the recombinant vaccine, the lambs were still directing IgG response to non-protective epitopes.  Perhaps additional studies focused on developing chimeric antigens using quantum vaccinomics is needed.”
Line 842 – change to: “that is crucial…”
Line 844 – change to “The VirB10 antigen…”
Response 1: We have corrected everything according to the suggestions.

Comments 2: I had suggested previously that in the summary statement the authors provide some overarching comment on the need to continue strong efforts for vaccines against these diseases.  As an example, I would offer: “The steadily increasing impact of tick-borne diseases on human and animal health and the economic factors on healthcare and agricultural productivity provide strong motivation for a much increased investment into research and development of a number of vaccines.  The current technical and financial challenges to create next-generation, efficacious products are well-recognized. Despite some progress as discussed here, steady advances in systems biology understanding of host-pathogen interactions along with bioinformatic-assisted approaches to vaccine development can potentially overcome these challenges.”
Response 2: We added with minor changes: 

The steadily increasing impact of tick-borne diseases on human and animal health and the economic factors on healthcare and agricultural productivity provide strong motivation for a much increased investment into research and development of a number of vaccines.  The current technical and financial challenges to create next-generation, efficacious products are well-recognized.  Despite those challenges, steady advances in systems biology understanding of host-pathogen interactions, along with bioinformatic-assisted approaches to vaccine development, can potentially overcome these challenges eventually.